# Effects of Structured and Unstructured Physical Activity on Gross Motor Skills in Preschool Students to Promote Sustainability in the Physical Education Classroom

Rosita Abusleme-Allimant [1], Juan Hurtado-Almonacid [1], Tomás Reyes-Amigo [2], Rodrigo Yáñez-Sepúlveda [3], Guillermo Cortés-Roco [4], Patricio Arroyo-Jofré [5] and Jacqueline Páez-Herrera [1,*]

1   Escuela de Educación Física, Pontificia Universidad Católica de Valparaíso, Valparaíso 2340000, Chile; juan.hurtado@pucv.cl (J.H.-A.)
2   Observatorio de Ciencias de la Actividad Física, Departamento de Ciencias de la Actividad Física, Universidad de Playa Ancha, Valparaíso 2340000, Chile
3   Faculty of Education and Social Sciences, Universidad Andres Bello, Viña del Mar 2520000, Chile
4   Escuela de Educación, Entrenador Deportivo, Universidad de Viña del Mar, Viña del Mar 2572007, Chile
5   Escuela Pedagogía en Educación Física, Universidad San Sebastian, Santiago 8420507, Chile
*   Correspondence: jacqueline.paez@pucv.cl

**Abstract:** Basic motor skills are the basis for the formation and execution of movements that will be utilized throughout an individual's lifetime, thus promoting their involvement and continued participation in physical activity. (1) Background: This study aimed to assess the impact of a physical education program, based on a model of structured and unstructured physical activity, on the motor development of kindergarten students at a private school for girls in Con Con, Chile. (2) Methods: Thirty-four female students were divided into two groups, one participated in structured physical activity and the other in unstructured physical activity, and both groups then underwent a 12-week intervention. The Test of Gross Motor Development-2 (TGMD-2) was utilized to evaluate motor behaviors, and the data were analyzed using descriptive statistics and relative frequencies. The Wilcoxon test was used to compare differences at the beginning and end of the intervention, while the Whitney–Mann U test was used to determine differences between groups. (3) Results: Statistically significant differences were observed in the overall group when comparing the start and end of the intervention for total motor development ($p = 0.001$), locomotion skills ($p = 0.018$), and object control ($p = 0.001$). However, no significant differences were found between the two types of intervention activities. (4) Conclusions: This study suggests that both structured and unstructured physical activity interventions enhance overall motor development, particularly in the dimensions of locomotion and object control. The results indicate that unstructured physical activity interventions may lead to better outcomes in motor development tests compared to structured interventions.

**Keywords:** motor skills; sustainable physical education; preschooler

## 1. Introduction

Ensuring a healthy life and at the same time promoting well-being for everyone at different ages is an essential element of Sustainable Development Goal number three, "Health and Wellbeing", proposed in the 2030 Agenda for Sustainable Development [1]. In this line, physical activity, defined as any bodily movement caused by muscles and causing energy expenditure [2], is essential for achieving the purposes of sustainability, as it is a strategy that can be implemented to protect people and the planet [3] as well as an action that would allow combating negative and destructive influences on society. Thus, the practice of physical activity can have positive effects since its influence can impact several generations in order to make them active and healthy people [3]. Thus, the regular practice of physical activity throughout life is a factor that directly influences the maintenance of an

individual's physical and mental health, contributes to weight control, and improves the quality of life and well-being. It is also an effective strategy for the treatment or prevention of chronic conditions [4].

In this sense, it is recognized that childhood is a fundamental stage to promote the acquisition of healthy habits, practice physical activity, and ultimately achieve optimal development [5–7], which allows an adult life free of diseases physically, organically, and cognitively [8].

In turn, the practice of physical activity and sports throughout life depends to a large extent on the construction of movement during childhood, with the development of basic motor skills [9,10]. The development of motor skills provides a series of benefits for children since they are a prerequisite for participation in motor activities, which in turn strongly benefit health [11–13].

Motor development refers to the changes in motor behavior that occur over the lifespan and is a result of the interplay between genetics, maturation, past experiences, environment, and new motor activities [14–17]. It is widely recognized that an adequate level of motor development in children is fundamental to developing specialized movement sequences for coping with daily life tasks and for organized and unorganized physical activities. Therefore, primary school is an essential stage for the development and improvement of motor skills [18].

In addition to this, during the preschool stage, it is possible to observe considerable progress in the development of basic motor and cognitive skills, as a result of maturation and learning capacity. This is why the preschool stage is essential to stimulate the motor development of children [19].

Basic motor skills are the fundamental structure upon which more complex motor responses are built, and the acquisition of mature movement patterns allows children to better interact with their environment [20,21]. Locomotion skills, such as walking, running, jumping, dodging, and balance skills, and manipulative skills, such as catching, throwing, kicking, propelling, and spinning, directly influence adherence to physical activity [22].

There is a positive correlation between motor development and health indicators [18–20] as well as physical activity levels [12]. Therefore, motor development plays a crucial role in the prevention of non-communicable diseases. In addition, the practice of physical activity provides diverse motor experiences, which, during the first years of childhood, constitute an essential axis for the development of motor skills [21].

There is even evidence that high performance in the development of basic motor skills at age 6 may act as a moderator for declining levels of physical activity between 6 and 10 years of age [22–26]. Similarly, evidence indicates that the low frequency of physical activity in individuals aged 5 to 17 years is partly explained by the low early acquisition of basic motor skills, which decreases the chances of participation in more complex motor activities [27–29].

In Chile, motor development levels are categorized as age-appropriate, under-age, or very poor [15,30,31]. Evidence indicates that one of the main problems in this situation is the lack of effective physical education classes, characterized by a reduced number of hours per week and a lack of specialized teachers. In addition to the above, a low weekly frequency of physical activity increases the gaps between children who have fewer and those who have more opportunities for motor practice [32], reflecting the inequity that can exist even in matters of motor stimulation.

Given the importance of motor development, physical education plays a crucial role in promoting healthy lifestyles and preventing chronic diseases in adulthood [33,34]. Thus, sustainable physical education (EFpODS) promotes the development of learning experiences that contribute to the capacity of human beings to be sustainable, developing an environment that is sustainable from economic, social, and environmental points of view [35]. Its role is fundamental since it constitutes one of the curricular spaces in a school in which it is possible to educate for the achievement of sustainable development objectives. Even UNESCO recognizes the practice of physical activity and physical exercise

as a direct strategy to promote health and well-being in schools. In this sense, motor skills can be developed using structured physical activity practice, which is teacher-directed and highly organized, or unstructured physical activity practice, which is peer-directed and unorganized play [36–38]. Structured physical activity practice, characterized by explicit instructions and constant feedback, focuses on immediate correction and leads to greater satisfaction and ownership [33,36–40]. In turn, physical education, with the realization of physical activity in structured contexts, leads to a greater sense of belonging [39] and offers the right amount of practice required for health improvement [41]. Structured practice promotes the development of basic motor skills in a linear fashion, considering a progression of skills from simple to complex and focusing primarily on the effectiveness of movement [42]. In turn, it is characterized by an emphasis on feedback, adequate organization of space, and didactic material. It is also called deliberate preparation [43]. Structured physical activity is predominant in physical education classes as well as in out-of-school activities [44].

Unstructured physical activity practice, in which the conditions of performance, rules, and pace are not determined, provides an opportunity for children to develop decision-making, communication, and problem-solving skills [45–47]. The contexts of physical activity practice in unstructured settings provide alternative and autonomous experiences [46], with the characteristic of exploration and with the intention of improving communication and self-management skills [33]. Similarly, they allow improving skills based on decision-making, communication, and problem-solving since, in a more autonomous practice environment, children must manipulate their environment to achieve positive results and maximize their satisfaction [46,47]. This type of class develops basic motor skills in a non-linear way, thus favoring creativity and interactions with peers and the environment, increasing the possibilities of movement exploration [42]. It is also characterized by the free disposition of resources and didactic equipment in the gymnasium or playground. According to the evidence, this type of physical activity results in a decrease in sedentary behavior and an increase in the amount and intensity of physical activity [48].

In 2015, the United Nations Educational, Scientific, and Cultural Organization emphasized that physical education is the sole subject responsible for promoting physical activity and gross motor development skills in children [49]. Physical education offers a vast array of pedagogical resources [50,51]. Several studies have shown that physical education classes positively affect children's development when they are involved in playful situations and paired and collaborative activities, and when teachers interact with them during the class [52].

Given the significant role that physical activity and motor development play in overall healthy development [53–56] and their positive correlation [5,12,57], it is crucial to not only incorporate practices that enhance motor skills and health but also to implement intentional, planned, relevant, and effective pedagogical proposals [58]. This will help to identify the most effective way to contribute to the comprehensive development of preschoolers.

Therefore, exploring educational strategies that encompass a range of physical education contexts is crucial for offering insight into how teachers and educators can optimally enhance health, motor learning, and development in early childhood. Likewise, the promotion of positive experiences from a physical education class is essential to achieve greater adherence to the practice of physical activity in the future and to reduce the anxiety and stress caused by daily life activities [35].

This study aims to assess the impact of a physical education program, based on a model of structured and unstructured physical activity, on the motor development of kindergarten students at a private school for girls in Con Con, Chile.

## 2. Materials and Methods

### 2.1. Participants

A total of 34 girls with an average age of 6.28 years participated in this study (in the structured group, the average age was 5.9 with SD = 0.3; in the unstructured group,

the average age was 6.0 with SD = 0.36), which lasted for a period of 12 classes (or three months). The participants in this study were enrolled in one of the two preschool classes at a private school in Con Con, Chile. The inclusion criteria for selecting participants included having a minimum class attendance of 70%, as well as not having a pathological condition that would prevent physical activity. Participants with lower attendance were excluded.

The participants were then randomly divided into two intervention groups using a computer-generated list: one group participated in structured physical activity (AFES) (17 students), while the other participated in unstructured physical activity (AFNOES) (17 students). The post hoc sample calculation was performed using the G—Power version 3.1 program. For this purpose, the Wilcoxon or Mann–Whitney test was used for two groups, corresponding to the *t*-test family. The sample size was 17 participants for the AFES group and 17 participants for the AFNOES group. With an alpha of 0.05 and an effect size of 0.38, the power (1-β err prob) was identified as 0.28.

The application of the instruments was conducted in accordance with the ethical principles for human research as outlined in the Declaration of Helsinki (World Medical Association, 2013) and the procedural and documentation suggestions of the Research Department of Pontificia Universidad Católica de Valparaíso through its Scientific and Bioethical Ethical Committee (BIOEPUCV-H-456-2021). The authorities of the educational establishment were consulted for authorization, and informed consent was obtained from the parents and/or guardians, which explained the objectives and scope of this study before their daughter's participation was approved. Throughout the development of this research, as well as in the course of writing this manuscript, all practices associated with research misconduct were avoided [59].

### 2.2. Design and Intervention

This study is a quasi-experimental study with a quantitative approach [58,60]. The participants were divided into two groups, each receiving one 45 min class per week. An 8 min motor activation period and a 7 min period for the conclusion of each class were allocated, leaving 30 min for the core of the class. Group one participated in physical education classes with structured physical activity, which involved tasks such as games with teacher-defined rules, circuit training, guided activities monitored and corrected by the teacher, and various motor tasks such as throwing a ball into a hoop, receiving a ball without displacement, running through obstacle courses, and movement imitation games. The activities performed by group one (structured physical activity) were led by one of the school's physical education teachers who was previously trained in class structure. Group two participated in physical education classes with unstructured physical activity, which involved tasks such as self-directed games and activities and free access to materials for the students. The activities performed by group two (unstructured physical activity) were directed by the physical education teacher in charge of this research work. Motor tasks for this group included free play with the various materials provided. In both types of classes, there was a closing of the class that consisted of tidying up the physical space, putting away the materials used in the session, and questions about the development of the class.

Both groups utilized sports initiation balls, cones, hurdles, Swedish benches, hoops, mats, tennis balls, and music equipment. The participants underwent pre- and post-intervention assessments after 12 classes. The TGMD-2 test [14], validated in Chile [61], was used to evaluate motor development. The test was administered by two physical education teachers at the educational establishment, who had experience in motor development measurements applicable to the TGMD-2, as well as by the person in charge of this research project. The test assessed 12 motor skills divided into two subtests: locomotion skills and object manipulation. Each task was scored based on efficacy and performance, with a score of one awarded for correct execution and zero otherwise. The assessments took place in a flat, obstacle-free space, and the participants wore comfortable clothing. The locomotion tests were performed individually and in the following order: running, galloping, hopping on one foot, jumping over an object, horizontal jumping, and lateral

displacement. The participants performed two attempts per test and received a score ranging from 0 to 2 points. Based on the total score and age/sex of each participant, motor development was categorized into seven categories: very poor (<70), poor (70–79), low-average (80–89), average (90–110), above-average (111–112), superior (121–130), and very-superior (>130) [14–61].

As an example, a card showing the design of the physical education classes with structured and unstructured physical activity is presented in Appendix A.

### 2.3. Recording Information and Statistical Analyses

Data were recorded using a TGMD-2 spreadsheet and organized with Microsoft Excel. Statistical analyses were performed with SPSS version 27. To test the normality of the data, the Shapiro–Wilk test (for samples of up to 50 data) was used, which determined that the data did not follow a normal distribution. The data were described using frequencies to group the number of girls at each level of motor development (very poor, poor, low-average, average, above-average, superior, and very-superior). This was performed for the total group, then for the structured physical activity group, and then for the unstructured physical activity group.

The analysis and nonparametric Wilcoxon tests were used to examine differences between the baseline and final measurements, and the Mann–Whitney U test was used to compare the structured and unstructured physical activity groups. Statistical significance was established at a 95% confidence level with a 5% margin of error (significant $p$-value < 0.05).

### 3. Results

The results obtained for the locomotion, manipulation, and TGMD-2 scores before and after the intervention for the total, structured, and unstructured groups are presented in Table 1.

**Table 1.** Gross motor skills before and after the intervention for the structured and unstructured physical activity (PA) groups.

| Test | Structured PA Group | | Unstructured PA Group | | $p$-Value [a] |
|---|---|---|---|---|---|
| | Before | After | Before | After | |
| TGMD-2 Score | 69.1 ± 8.1 | 86.2 ± 4.3 | 70.4 ± 6.2 | 83.9 ± 7.4 | 0.278 |
| Locomotor Score | 39.2 ± 6.6 | 44.1 ± 2.8 | 39.5 ± 5.2 | 43.8 ± 3.6 | 0.793 |
| Object Control Score | 29.8 ± 4.1 | 42.2 ± 3.5 | 30.8 ± 3.9 | 40.2 ± 5.0 | 0.184 |

Data are mean ± SD. [a] An independent sample $t$-test was used for comparison between groups. PA: physical activity.

In the structured and unstructured physical activity groups, the participants scored a higher locomotor score, object control score, and TGMD-2 score after the intervention. It was observed, for all the motor skills, that the structured physical activity group improved more than the unstructured physical activity group; however, the observed difference was not statistically significant for the TGMD-2 score ($p$-value = 0.278), locomotor score ($p$-value = 0.793), or object control score ($p$-value = 0.184).

The results of this study are presented in Table 2 and are categorized by structured and unstructured physical activity groups.

**Table 2.** Locomotion motor development, according to the total group and the structured and unstructured physical activity groups.

| Development Level | Group Total | | | PA Group Structured | | PA Group Unstructured | | *p*-Value ** |
|---|---|---|---|---|---|---|---|---|
| | **Before** | **After** | ***p*-Value *** | **Before** | **After** | **Before** | **After** | |
| Very Poor | 0 (0.0) | 0 (0.0) | | 0 (0.0) | 0 (0.0) | 0 (0.0) | 0 (0.0) | |
| Poor | 0 (0.0) | 0 (0.0) | | 0 (0.0) | 0 (0.0) | 0 (0.0) | 0 (0.0) | |
| Below-Average | 2 (5.9) | 0 (0.0) | | 0 (0.0) | 0 (0.0) | 2 (11.8) | 0 (0.0) | |
| Average | 19 (55.9) | 11 (32.4) | 0.018 *** | 11 (64.7) | 5 (29.4) | 8 (47.1) | 6 (35.3) | 0.642 |
| About Average | 3 (8.8) | 8 (23.5) | | 2 (11.8) | 4 (23.5) | 1 (5.9) | 4 (23.5) | |
| Superior | 8 (23.5) | 10 (29.4) | | 3 (17.6) | 5 (29.4) | 5 (29.4) | 5 (29.4) | |
| Very-Superior | 2 (5.9) | 5 (14.7) | | 1 (5.9) | 3 (17.6) | 1 (5.9) | 2 (11.8) | |

Data are shown in n (%); PA: physical activity; *: the Wilcoxon test was used to compare the total group before and after the intervention; **: the Mann–Whitney U test was used to compare groups after the intervention; ***: *p*-value < 0.05.

The level of motor development in locomotion skills, pre- and post-intervention, was analyzed. The results showed that prior to the intervention, the total group of students displayed categories ranging from low-average (5.9%) to superior and very-superior (29.4%). However, after the intervention, no students exhibited a low-average level. Additionally, the number of students with an average level decreased by 23.5 percentage points, while the number of students with an above-average level increased by 14.7 percentage points, and the superior and very-superior categories increased by 14.7 percentage points (from 29.4% to 44.1%). This difference between pre- and post-intervention was found to be statistically significant, with a *p*-value of 0.018. A comparison between the structured and unstructured physical activity groups revealed similar patterns in the development of locomotion skills, with the majority of students falling into the average, above-average, and superior categories. No statistically significant difference was observed between the two groups.

The evolution of manipulative skills before and after the intervention is presented in Table 3. Prior to the intervention, 5.8% of the students in the total group were categorized as having poor or low-average skills, while 91.2% were average and only one student was above average. After the intervention, no students were categorized as having poor or low-average skills, while the number of average students decreased by 73.6% points to six students, and the majority (82.3%) showed levels of above-average, superior, and very-superior, with an increase of 79.4%. The difference was statically significant with a *p*-value of 0.001. The comparison between the structured and unstructured physical activity groups showed similar patterns in manipulative skill development, with no significant difference between the groups.

Table 4 displays the pre- and post-intervention gross motor development. Before the intervention, 2.9% of the students were categorized as low-average, 70.6% as average, 14.7% as above-average, and 11.8% as superior. After the intervention, no students were categorized as low-average, the number of average students decreased by 61.8%, the number of above-average students increased by 8.8 percentage points, and the number of superior and very-superior students increased by 32.3%, with 23.5% now in these categories. The difference between pre- and post-intervention gross motor development was statistically significant with a *p*-value of 0.001.

A comparison between the structured and unstructured physical activity groups showed similar patterns in gross motor development, with no significant difference between the groups.

**Table 3.** Manipulation motor development according to the total group and the structured and unstructured physical activity groups.

| Development Level | Group Total | | | PA Group Structured | | PA Group Unstructured | | p-Value ** |
|---|---|---|---|---|---|---|---|---|
| | **Before** | **After** | **p-Value *** | **Before** | **After** | **Before** | **After** | |
| Very Poor | 0 (0.0) | 0 (0.0) | | 0 (0.0) | 0 (0.0) | 0 (0.0) | 0 (0.0) | |
| Poor | 1 (2.9) | 0 (0.0) | | 1 (5.9) | 0 (0.0) | 0 (0.0) | 0 (0.0) | |
| Below-Average | 1 (2.9) | 0 (0.0) | | 0 (0.0) | 0 (0.0) | 1 (5.9) | 0 (0.0) | |
| Average | 31 (91.2) | 6 (17.6) | 0.001 *** | 16 (94.1) | 2 (11.8) | 15 (88.2) | 4 (23.5) | 0.234 |
| About Average | 1 (2.9) | 17 (50.0) | | 0 (0.0) | 8 (47.1) | 1 (5.9) | 9 (52.9) | |
| Superior | 0 (0.0) | 3 (8.8) | | 0 (0.0) | 2 (11.8) | 0 (0.0) | 1 (5.9) | |
| Very-Superior | 0 (0.0) | 8 (23.5) | | 0 (0.0) | 5 (29.4) | 0 (0.0) | 3 (17.6) | |

Data are shown in n (%); PA: physical activity; *: the Wilcoxon test was used to compare the total group before and after the intervention; **: the Mann–Whitney U test was used to compare groups after the intervention; ***: $p$-value < 0.05.

**Table 4.** Gross motor development according to the total group and the structured and unstructured physical activity groups.

| Development Level | Group Total | | | PA Group Structured | | PA Group Unstructured | | p-Value ** |
|---|---|---|---|---|---|---|---|---|
| | **Before** | **After** | **p-Value *** | **Before** | **After** | **Before** | **After** | |
| Very Poor | 0 (0.0) | 0 (0.0) | | 0 (0.0) | 0 (0.0) | 0 (0.0) | 0 (0.0) | |
| Poor | 0 (0.0) | 0 (0.0) | | 0 (0.0) | 0 (0.0) | 0 (0.0) | 0 (0.0) | |
| Below-Average | 1 (2.9) | 0 (0.0) | | 1 (5.9) | 0 (0.0) | 0 (0.0) | 0 (0.0) | |
| Average | 24 (70.6) | 3 (8.8) | 0.001 *** | 10 (58.8) | 1 (5.9) | 14 (82.4) | 2 (11.8) | 0.465 |
| About Average | 5 (14.7) | 8 (23.5) | | 4 (23.5) | 4 (23.5) | 1 (5.9) | 4 (23.5) | |
| Superior | 4 (11.8) | 15 (44.1) | | 2 (11.8) | 7 (41.2) | 2 (11.8) | 8 (47.1) | |
| Very-Superior | 0 (0.0) | 8 (23.5) | | 0 (0.0) | 5 (29.4) | 0 (0.0) | 3 (17.6) | |

Data are shown in n (%); PA: physical activity; *: the Wilcoxon test was used to compare the total group before and after the intervention; **: the Mann–Whitney U test was used to compare groups after the intervention; ***: $p$-value < 0.05.

## 4. Discussion

The purpose of this study was to examine the impact of a physical education program, incorporating both structured and unstructured physical activities, on the motor development of kindergarten students at a private school for girls.

The results of this study indicate that girls belonging to both groups (structured physical activity and unstructured physical activity) had improvements in the motor skills of locomotion, object control, and total motor development. However, there were no statistically significant differences when comparing motor development and its dimensions between the two groups.

In contrast to our results, a program consisting of sports, games, and recreational activities in a physical education class resulted in significant improvements in the Test of Gross Motor Development-2 (TGMD-2) for the children who participated. In contrast, those who took part in a gymnastics class or a physical activity routine did not show significant changes after eight weeks of intervention [62]. In another study, after a four-week intervention, there was only an increase in the scores for basic motor skills in the intervention group who participated in sessions composed of games that stimulate the development of motor skills [22]. These differences are mainly explained by the weekly frequency of practice and the total time of the intervention [60].

As in our study, there is similar evidence as no significant differences in motor development were reported as a function of the type of physical activity (structured and unstructured) [38,63].

However, an intervention based on structured, four-week classes for children aged 8 to 10 years, with a weekly frequency of two 30 min sessions, reported significant improvements in the mastery and efficiency of basic motor skills including hopping on one foot, hopping with the feet together, throwing a ball over the shoulder, and running over a distance of 20 m [23]. There is evidence indicating that when treated in a structured manner, with a linear progression of skills and with clear movement restrictions, for example, limiting the spatial position, the temporal sequence of motor actions that compose the skills and system of rules of the motor tasks favor the development of motor efficiency and competence [42].

No statistically significant differences were observed between both groups post-intervention for the locomotion tests based on the type of structured and unstructured activities. However, in a different study, children who participated in a program of traditional games performed better on locomotion tests compared to those who engaged in daily activities at school [64]. The results of our study differ because, for the object control tests, both groups of girls who participated in structured and unstructured activities showed improvement in their performance. Yet, no statistically significant differences were observed between the two groups. The evidence indicates that interventions in the physical education classroom, whether structured or unstructured, report considerable improvements in those motor skills related to object control. Its benefits lie in the improvement of visuomotor coordination and overall body control [9].

Evidence suggests that a physical education class intervention focusing on selected exercises for motor development in third-grade girls produced significant differences in favor of the experimental group on object control tests [65].

The results of the present study are in contrast to previous findings reported in other studies that suggest structured physical activity practices have a statistically significant impact on the promotion and development of motor skills in preschool children [65–68]. However, the composition of physical activities for preschool children, including teacher influence, teaching methods, intensity, and frequency of activity, can significantly impact their development [9]. It has been suggested that the organization and amount of intervention play a role in the results of motor development [24]. Although previous research suggests that interventions should be performed on a frequent basis to enhance motor development [68], the current study shows that both structured and unstructured physical activity can lead to improvements in motor development.

In this sense, the results of this study are supported by evidence that indicates that both a structured and an unstructured approach to physical education classes achieve benefits for the motor development of preschoolers. The weekly frequency, the type of content, and the characteristics of the teachers are fundamental to the effect of the intervention [9].

It is recommended that both structured and unstructured physical activity be offered to young girls in early childhood, rather than only one form. Providing unstructured physical activity opportunities similar to recreation and free play may increase participation in structured physical education contexts [33], potentially leading to enhanced basic motor skills such as manipulation and locomotion. Additionally, the stimulation of motor development, independent of intervention effects, is in opposition to the maturation theory, which states that the acquisition of skills depends more on internal factors than on external or environmental factors [38,67,69,70].

Regarding the limitations of this study, it is possible to point out the small number of participants in the intervention. Another limitation is the exclusive participation of girls and not of boys. Finally, we consider that it is important to include a control and experimental group, which would allow us to compare the effect of interventions of this nature.

## 5. Conclusions

It is concluded that the effect of the implementation of 12 classes designed under the model of structured and unstructured physical activity practice produces an improvement in the development of motor skills including locomotion, manipulation, and gross motor development. On the other hand, no statistically significant difference was reported in motor development between the groups participating in structured and unstructured physical activity practice. It is suggested to continue investigating the effect that the incorporation of structured and unstructured practices in physical education classes has on motor development. Likewise, inquiring about those teaching strategies and learning activities that physical education teachers implement in the physical education class to favor motor development is an opportunity to expand the evidence on this topic.

A physical education class has a high potential to promote the acquisition of sustainability competencies in children and youth. Therefore, it is essential to reflect on how the learning experiences and teaching methods implemented in physical education classes contribute to the development of students with a sustainable awareness that allows them to be active agents in today's society.

Among the strengths of this study, we identified richness from the implementation of structured and unstructured physical activity and its benefits in motor development. The learning proposals that shape the development of the classes, under both ways of approaching physical activity, also constitute a strength in the intervention proposal.

In terms of the future lines of research that have been generated from this study, it is mainly suggested to investigate the characteristics of physical education classes and the positions used by teachers in terms of their interaction with students and how these can be generated.

**Author Contributions:** Conceptualization, R.A.-A. and J.P.-H.; methodology, R.A.-A. and J.P.-H.; software, R.A.-A. and J.P.-H.; validation, J.P.-H., R.Y.-S. and T.R.-A.; formal analysis, R.A.-A.; investigation, R.A.-A., J.P.-H. and J.H.-A.; resources, R.A.-A. and J.P.-H.; data curation, R.A.-A. and R.Y.-S.; writing—original draft preparation, R.A.-A., J.P.-H. and J.H.-A.; writing—review and editing, J.P.-H., J.H.-A., T.R.-A. and R.Y.-S.; visualization, G.C.-R. and P.A.-J.; supervision, G.C.-R. and P.A.-J.; project administration J.P.-H.; funding acquisition, R.A.-A. All authors have read and agreed to the published version of the manuscript.

**Funding:** This research received no external funding.

**Institutional Review Board Statement:** This study was conducted in accordance with the Declaration of Helsinki and approved by the Ethics Committee of Pontificia Universidad Católica de Valparaíso (BIOEPUCV-H-456-2021) for studies involving humans.

**Informed Consent Statement:** Informed consent was obtained from all subjects involved in this study.

**Data Availability Statement:** The data supporting the results of this study are available from the corresponding authors.

**Acknowledgments:** We would like to thank St Margaret's British School for Girls and the parents and guardians for authorizing this study. We would also like to thank the Master's program in Physical Activity for Health of the School of Physical Education of the Pontificia Universidad Católica de Valparaíso and the National Agency for Research and Development (ANID) for encouraging this study.

**Conflicts of Interest:** The authors declare no conflict of interest.

**Appendix A**

| Class Design | |
|---|---|
| **Subject: Sports** | **Course: Kinder** |
| **Learning Unit** | **Motor Development** |
| **Program Learning Objective** | Solve practical challenges while maintaining control, balance, and coordination by combining various movements, postures, and movements such as throwing and catching, moving on inclined planes, and following rhythms in a variety of games. <br> Coordinate their motor skills by practicing strength, endurance, and traction postures and movements such as throwing rope, carrying objects, and using implements in every day and play situations. |
| **Class Learning Objective** | Execute basic motor skills of manipulation, locomotion, and stability in a controlled and coordinated manner in a variety of games, courses, and circuits. |

| Sequence of Learning Activities | | |
|---|---|---|
| **Learning Activities** | **Resources and Materials** | **Evaluation Criteria and Indicators** |
| Group Structured physical activity (games and tasks intentionally directed by the teacher, workspaces, well-defined and organized moments). <br><br> - Beginning (8 min): Attendance register and group order. As an activation, the game "The floor is lava" is played. The students must free themselves from the "lava" by climbing on the materials arranged in the space in order to avoid being trapped by the lava when it "appears" (as indicated by the teacher). The materials are fixed, and they do not change position as the game develops. Materials of similar height and dimensions are selected. <br> - Development (30 min): For the development of the class, the students are distributed in three groups, each with five to six members, in which they carry out relay tasks. <br> Relay 1: The whole group executes the tasks individually and successively one after the other. The first task is to walk on ropes (three) separated by a distance of 3 m. <br> Relay 2: Each of the girls must move (back and forth) bouncing a ball for a distance of 10 m. <br> Relay 3: Each of the girls must move by rolling a ball on the ground for a distance of 10 m. <br> Relay 4: Each of the girls must jump over four obstacles (20 cm high) continuously for a distance of 10 m (round trip). <br> There is constant monitoring and correction of the execution of the tasks. The obstacles, distance, and amount of material are not modified throughout the session. <br><br> - Closing (7 min): Return to calm, order materials, and verify what was learned based on the following question: What is more difficult and what is easier? | Balls <br> Mats <br> Cones <br> Hoops <br> Hurdles | Executes basic manipulative motor skills in a controlled and coordinated manner. <br> Indicators: <br><br> 1. When throwing, body weight is transferred by stepping forward with the foot opposite the throwing hand. <br> 2. When bouncing, the hand contacts the ball at waist height. <br> 3. Maintains consecutive bounces of the ball without letting it escape. <br> 4. When receiving the ball, extends the arms to reach the ball. <br> 5. Catches the ball with hands. <br> 6. Strikes the ball with the toe of the dominant foot. <br> 7. When rolling the ball, releases the ball close to the ground. <br> 8. Coordinates his/her body segments while moving. <br> 9. When jumping, he/she uses the impulse of his/her arms. <br> 10. When jumping, projects the body in the indicated direction. |

| Class Design | |
|---|---|
| **Subject: Sports** | **Course: Kinder** |
| **Learning Unit** | **Motor Development** |
| **Program Learning Objective** | Solve practical challenges while maintaining control, balance, and coordination by combining various movements, postures, and movements such as throwing and catching, moving on inclined planes, and following rhythms in a variety of games.<br>Coordinate their motor skills by practicing strength, endurance, and traction postures and movements such as throwing rope, carrying objects, and using implements in every day and play situations. |
| **Class Learning Objective** | Execute basic motor skills of manipulation, locomotion, and stability in a controlled and coordinated manner in a variety of games, courses, and circuits. |

| Sequences of Learning Activities | | |
|---|---|---|
| **Learning Activities** | **Resources and Materials** | **Evaluation Criteria and Indicators** |
| Group PA unstructured (self-selected free games and tasks with less defined spaces and moments)<br><br>- Beginning (8 min): A time of 8 min is available for activating the students. For this, music is played for the development of the class, and the students are given instructions to "move freely through the space, to the rhythm of the music, in the way they like".<br>- Development (30 min): Game of the students' free choice.<br><br>Arrangement of materials of different weights, sizes, and colors to generate a space for the self-selection of tasks and games. On this occasion, sports initiation balls, ropes, initiation hurdles, sponge balls, and tennis balls, among others, are made available.<br>The materials are not distributed in an established order, nor are they organized by type of equipment.<br>There is no constant monitoring and correction of the execution of the tasks.<br>Closing (7 min): Order the materials and verify what was learned based on the following question: What is more difficult and what is easier? | Balls<br>Mats<br>Cones<br>Hoops<br>Hurdles | Executes basic manipulative motor skills in a controlled and coordinated manner.<br>Indicators:<br><br>1. When throwing, body weight is transferred by stepping forward with the foot opposite the throwing hand.<br>2. When bouncing, the hand contacts the ball at waist height.<br>3. Maintains consecutive bounces of the ball without letting it escape.<br>4. When receiving the ball, extends the arms to reach the ball.<br>5. Catches the ball with hands.<br>6. Strikes the ball with the toe of the dominant foot.<br>7. When rolling the ball, releases the ball close to the ground.<br>8. Coordinates his/her body segments while moving.<br>9. When jumping, he/she uses the impulse of his/her arms.<br>10. When jumping, projects the body in the indicated direction. |

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
