# Peer review of "Effects of Structured and Unstructured Physical Activity on Gross Motor Skills in Preschool Students to Promote Sustainability in the Physical Education Classroom"

_sustainability, doi:10.3390/su151310167_

Round 1
Reviewer 1 Report
It will continue to review the following sections:
1. Methodology must explain the research design, type of research, as well as the procedure of the intervention. When recording the information, they state that the Wilcoxon test was used to examine the differences between the initial and final measurements. It would be necessary to indicate if there was a pretest and posttest, which one was carried out. They should improve the study methodology.
2. Incorporate a section on study limitations.
3. New lines of research have been generated from the study.
Author Response
Dear Reviewer
We would like to thank you for your comments, which have allowed us to improve the article.
Please see attached file
Sincerely

Reviewer 2 Report
The name of the school should not be indicated; The pedagogically unstructured proposal must be differentiated; The sample normality test should be indicated (Shapiro Wilk since that indicates the statistical test of comparison of means; How is it explained that the changes with statistically significant differences are due to stimulation and not to the process of maturation and growth? It seems to me that a control group without intervention was missing; is a relevant topic for the analysis of appointments (9-12- and 14; Informed consent should not be part of the inclusion criteria; It is convenient to incorporate a standard file of the procedure of both types of sessions or classes. This is important as for other results verification studies the same model or format should be used, for example start phase, main part and end phase of the session; It is important to mention why Wilcoson was used and not Student's; the t for student is lowercase; It is necessary to incorporate the DOI or the URLs of each Bibliographic reference.Author Response
Dear Reviewer
We would like to thank you for your comments, as they have been valuable for the improvement of the document. Please review the attached file.
Sincerely

Reviewer 3 Report
The study investigates On the Analysis of the impact of different physical education programs, structured or unstructured, on the health and motor development of preschool and school-age children, a Question is
There are still few studies that analyse the benefits of different approaches even with respect to the different teaching approach that teachers take in implementing this or that type of activity. In this sense, the study by Abusleme et al. although not original is relevant because studying its effects through an empirical study
The article has some critical issues for the part of the ricercar, but is good from the point of view of presenting the issue and theoretical background.
The research design is good overall even though the standard deviation of the sample age is not specified nor the class from which the children were selected.
The authors should specify:
- The standard deviation of the 'age of the sample
- The school classes of the pupils
the sample is limited in size and is only female, you could specify for what motive a group of only females was analysed
The data analysis was clearly and adequately prepared as well as the presentation of the results.
Other studies that have investigated the same question should be presented in the state of the art of research.
Conclusions should highlight the strengths and weaknesses of the research and future prospects.
Missing from the introduction is a description of studies that have investigated the same question.
The tables are clear in their representation of the results of the study.
The research design is good overall even though the standard deviation of the sample age is not specified nor the class from which the children were selected; moreover, the sample is limited in size and is only female.
The data analysis was clearly and adequately prepared as well as the presentation of the results.
Author Response
Dear Reviewer
We would like to thank you for your kindness in your correction, which has allowed us to improve the manuscript.
Please review the attached document, where we have given some answers to your request.
Sincerely

Reviewer 4 Report
Review Manuscript sustainability-2366189, entitled “Effects of Structured and Unstructured Physical Activity in Gross Motor Skills of Preschool Students.”
Journal: Sustainability
Dear Authors.
The study concerns an important problem of the impact of the physical education program, based on a model of structured and unstructured physical activity, on the motor development of kindergarten students. I have a couple of requests to be revised as stated below.
Comments:
1. In my opinion, the Introduction is too long.
Materials and Methods section
2. Inclusion and exclusion criteria: There is no information about the health of children. Whether there were any disease entities that constituted the exclusion criterion?
3. Were the children from one pre-school group?
4. Why classes were conducted only once a week?
5. Who conducted the activities within the program?
6. Who conducted the TGMD-2 test?
7. Categories of motor development should be clarified: How many points meant which level?
8. Sample size information is missing.
Results section
9. Low average (line 211) or below average (Table 1, 2, 3). Should be unified.
10. It is unnecessary to compare the level of development before and after the intervention. The aim was to compare the motor development depending on the forms of physical activity (structured vs unstructured). One table should show a comparison of motor development in both groups. It would be better if there was no difference between them. The next tables should show the differences between the groups in the areas of locomotion, manipulation and gross motor development.
11. There is no information about which skills have been improved. There is only a level of motor development.
Discussion
12. Lines 263-265: The results reveal that there was a significant improvement in the overall group’s locomotion, manipulative, and gross motor skills. This is obvious. The purpose of this study was different: to assess the impact of the physical education program, based on a model of structured and unstructured physical activity, on the motor development of kindergarten students.
Thank you very much. I think the article should be corrected and supplemented with missing information.
Author Response

(The authors gave the same response as above.)

Round 2
Reviewer 4 Report
Review Manuscript sustainability-2366189, entitled “Effects of Structured and Unstructured Physical Activity in Gross Motor Skills of Preschool Students to promote sustainability in the physical education classrom.”
Journal: Sustainability
Dear Authors.
Some changes have been made. Below are the unaddressed suggestions from the previous review
Comments:
11. In my opinion, the Introduction is too long.
Materials and Methods section
22. Point 4: Were they the children in a preschool group?
My question was: Were the children from one pre-school group?
33. Sample size information is missing.
44. Line 210: This study is quasi-experimental study with a quantitative approach. Not only quantitative but also qualitative. You converted quantitative data into motor development categories (lines 242-243). The parameter was described using basic descriptive statistics measurements, i.e., the percentage.
55. Lines 242-243: What are these categories based on? Reference may be added. If you created it yourself, the levels with 0% are difficult to analyze statistically.
Results section
66. It is unnecessary to compare the level of motor development before and after the intervention. The aim was to compare the motor development depending on the forms of physical activity (structured vs unstructured). In my opinion, the presentation of data for the total group should be removed. My tips for table titles and their contents:
Table 1. Gross motor skill before the intervention
This table should show whether there was any difference between the groups before the intervention
Table 2. Gross motor skill, before and after intervention, in Structured and Unstructured Physical Activity (PA) groups.
p should be calculated for each group (Structured and Unstructured) separately, comparing the results before and after the intervention
Table 3. Gross motor skill after the intervention
This table should show whether there was any difference between the groups after the intervention
77. Is there a need to show the levels of motor development? In order to replace quantitative data with qualitative data, it is better to add total motor development in the previous tables and show the total sum of points before and after the intervention in each group.
88. Statistical tests used should be checked. there is no information about tests comparing qualitative data, for example under Table 5.
99. Lines 317, 333, 349: p-value should be indicated separately for before and after intervention in unstructured physical activity practice group, and for before and after intervention in structured physical activity practice groups. You have one p value for the total group and the other for before and after for both groups together (structured and unstructured). I don't understand these calculations.
Discussion
110. Lines 355-357: The results reveal that there was a significant improvement in the overall group’s locomotion, manipulative, and gross motor skills. The purpose of this study was different: to assess the impact of the physical education program, based on a model of structured and unstructured physical activity, on the motor development of kindergarten students. Not overall group, but by comparing groups with different interventions.
111. Limitations should come before the Conclusions section. This is rather the last paragraph of the discussion.
Thank you very much. I think the article should be corrected and supplemented with missing information.
Author Response
Dear Reviewer
We would like to greet you and share with you the letter of response to each one of the points.
Grateful for your valuable comments, you are cordially dismissed,

Round 3
Reviewer 4 Report
Dear Authors.
Despite the limitations, I congratulate the authors for their work. I think the study should be accepted and published by the Sustainability.